# CodeStructEval: A Holistic Evaluation Framework of Code Structure Generation and Comprehension

## Abstract

As Large Language Models (LLMs) rapidly evolve and demonstrate strong performances in software engineering tasks, a growing number of researchers are focusing on evaluating LLMs' code generation capabilities. Different from previous benchmarks that primarily focus on evaluating LLMs' ability to generate sequential code from natural language requirements, we propose to assess their capabilities in generating and comprehending code structures. The two aspects represent a deeper, more fundamental understanding of program logic that better reflects the model's capacity for logical reasoning and structural awareness. Specifically, in the paper, we formally propose two tasks: CSG (**C**ode **S**tructure **G**eneration) and CSC (**C**ode **S**tructure **C**omprehension). The former requires LLMs to generate code structural information from given code, while the latter requires it to generate code from given code structural information. Then, we design a holistic evaluation framework called CodeStructEval to assess LLMs' code structure generation and comprehension capabilities. This programming language agnostic evaluation framework has three main parts: 1) data preprocessing, 2) model inference, and 3) automated evaluation. For evaluation metrics, we introduce SAR (**S**emantic **A**ccuracy **R**ate) and StAR (**St**ructure **A**ccuracy **R**ate) to assess LLM' output quality semantically and structurally, respectively. Then, using the CodeStructEval framework and the HumanEval seed dataset, we built a benchmark with 157 samples across three difficulty levels (Easy, Medium, Hard). At last, we use this benchmark to thoroughly evaluate the code structure generation and comprehension abilities of 18 mainstream LLMs. Our experimental results show that closed-source commercial LLMs demonstrate strong code structure generation and comprehension capabilities, while smaller open-source LLMs still have room for improvement.

## 1 Introduction

In recent years, with the continuous advancement of large language models (LLMs), more and more researchers are beginning to care about the performance of LLMs in software engineering, especially in the field of code: code completion (Jiang et al., 2024; Chen et al., 2021), program repair (Tang et al., 2024; Jain et al., 2024), program debugging (Tian et al., 2024; Yang et al., 2025), test case generation (Li & Yuan, 2024; Yang et al., 2024), and code optimization (Gong et al., 2025; Deng et al., 2025). Recent models such as CodeLlama (Rozière et al., 2024), Qwen2.5-Coder (Hui et al., 2024), and Claude-3.5-Sonnet (Anthropic, 2024) have shown promising results in code-related tasks and are being used to develop tools to help programmers write code more efficiently.

Unlike natural language, code typically adheres to rigid syntactic constructs. When developers employ LLMs for downstream tasks such as code completion and progam repair, the primary objective is to generate code that is immediately executable, readable, and tailored to human requirements (Austin et al., 2021; Liu et al., 2023; Xin & Reiss, 2017; Li et al., 2020). However, existing LLMs may encounter issues such as the generation of non-functional or syntactically incorrect code, which can be attributed to a lack of comprehension of code structure (Chen et al., 2025; Jesse et al., 2023; Kou et al., 2024; Mu et al., 2023). In addtion, existing researches show that providing code structure information during model training can effectively improve the performance of the model in

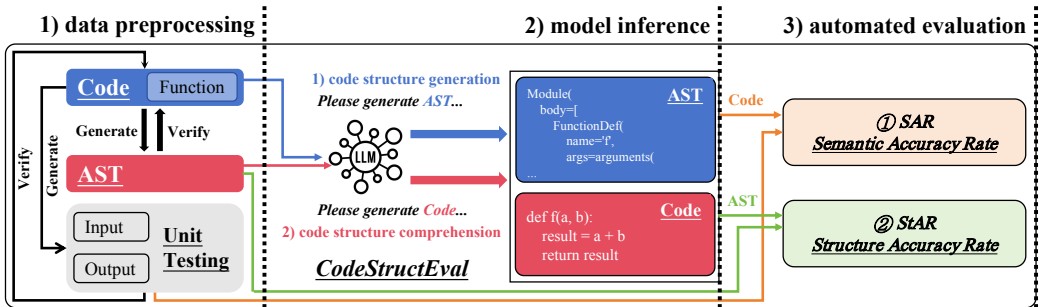

Figure 1: Schematic diagram of CodeStrcutEval evaluation framework.

various downstream tasks (Zhang et al., 2025; Gong et al., 2024; Wu et al., 2024). Researchers have made preliminary attempts to evaluate the ability of LLMs to understand code structure, but existing research faces various problems such as the lack of a systematic evaluation framework, single evaluation metric and reliance on manual evaluation Ma et al. (2024a;b). In response to the above research status, we makes the following efforts:

First, abstract syntax trees (AST) directly represent the structural information of the code in a tree-like hierarchical structure by stripping away unstructured grammatical details. Their language independence and automatic parsing features make them more suitable for evaluating the code structure generation or comprehension capabilities of LLMs. Therefore, leveraging AST, we formally propose two tasks: CSG (**C**ode **S**tructure **G**eneration) and CSC (**C**ode **S**tructure **C**omprehension). These tasks require the model to generate code structure information (AST) from the given code, and to generate code from the given code structure information (AST). Then, we propose a programming language agnostic evaluation framework called CodeStructEval (as shown in fig. 1). This framework encompasses three components: 1) data preprocessing, 2) model inference, and 3) automated evaluation. Specifically, given a seed dataset containing code, the CodeStructEval evaluation framework transforms it into a benchmark for evaluating the code structure generation and comprehension capabilities of LLMs. Regarding evaluation metrics, we propose SAR (**S**emantic **A**ccuracy **R**ate) and StAR (**St**ructure **A**ccuracy **R**ate), which assess the quality of the model's output semantically and structurally, respectively. Cases of CSG and CSC tasks are shown in fig. 4.

Then, guided by the CodeStructEval evaluation framework, we construct a benchmark based on the HumanEval (Chen et al., 2021) seed dataset. HumanEval is a handwritten code generation dataset, where each sample contains a verified executable code and available assertions. We first extract the code from the HumanEval samples and automatically generated AST data for each code segment. We then extract the assertions in the samples as our unit tests to calculate the corresponding evaluation metrics. After rigorous screening (runtime screening, complexity screening, and manual screening), we obtain a total of 157 evaluation samples. Finally, we classify the difficulty of each sample into "Easy" (63 in total), "Medium" (63 in total), and "Hard" (31 in total) based on its complexity (the token length of each sample).

Finally, we select 18 mainstream LLMs to thoroughly evaluate their code structure generation and comprehension capabilities. These models include both open-source and closed-source models, both base models and instruct models, both small-parameter version (1.5B) and large-parameter version (14B) of the same model series, and both general models and code-specific models. We also conduct a detailed quantitative correlation analysis, including the correlation between the model's scores on our benchmark and those on HumanEval, as well as the correlation within our benchmark. In addition, we also attempt to use Chain of Thought (Wei et al., 2023) and Few-Shot (Brown et al., 2020) techniques to improve the model's code structure generation and comprehension capabilities and obtained interesting findings.

In summary, our contributions are as follows:

- We formally propose two tasks: CSG (Code Structure Genearation) and CSC (Code Structure Comprehension). Based on these two tasks, we design an evaluation framework called

CodeStructEval and two evaluation metrics to comprehensively assess the code structure generation and comprehension capabilities.

- We construct a benchmark based on the CodeStructEval evaluation framework and the Humaneval seed dataset. After processing and filtering, the benchmark contains a total of 157 data, and is divided into three difficulty levels: Easy, Medium, and Hard according to the complexity of the samples.

- We conduct a thorough review of the code structure generation and comprehension capabilities of 18 mainstream LLMs, accompanied by detailed analytical experiments. We believe that the conclusions of the experiments will facilitate researchers' insights into the code structure generation and comprehension capabilities of LLMs.

## 2 RELATED WORK

### 2.1 LARGE LANGUAGE MODELS ON CODE

The application of pre-trained language models to code generation tasks has undergone a significant evolution. Early research primarily focused on adapting existing architectures, such as in Code-BERT (Feng et al., 2020) and GraphCodeBERT (Guo et al., 2021). These models were pre-trained on a bimodal mixture of natural language and code data, a strategy designed to enable them to comprehend the semantics and intrinsic structure of code. However, as model size and computational resources rapidly increased, the research focus experienced a pivotal shift. The paradigm moved away from adapting mixed-data models towards training massive, specialized foundation models almost exclusively on code. This newer generation of models, exemplified by CodeLlama (Rozière et al., 2024), CodeGemma (Team et al., 2024), and Qwen2.5-Coder (Hui et al., 2024), leverages vast repositories of source code to develop a deeper, more nuanced fluency in programming languages, significantly advancing the state of the art in automated code generation.

### 2.2 EVALUATION OF CODE COMPREHENSION ABILITY

The evaluation of a large language model's (LLM) code comprehension has moved beyond simple semantic understanding. Current methods now employ more sophisticated, multi-faceted benchmarks to assess a model's ability to reason, execute, and even predict the behavior of code. For example, LiveCodeBench (Jain et al., 2024) utilizes questions from popular competitive programming platforms like LeetCode, AtCoder, and Codeforces. It provides a comprehensive evaluation by testing a model's capacity for code execution, self-repair, and test output prediction. This approach offers a holistic view of a model's practical coding skills, simulating real-world problem-solving scenarios. CRUXEval (Gu et al., 2024) is another benchmark that delves into code reasoning. It's built around 800 Python functions with corresponding input-output pairs. This benchmark is divided into two subtasks: input prediction (CRUXEval-I) and output prediction (CRUXEval-O). By requiring models to predict both the inputs that would lead to a specific output and the outputs for a given input, CRUXEval effectively measures a model's ability to reason about and comprehend code logic. REval (Chen et al., 2024) provides a framework focused on runtime behavioral reasoning and incremental consistency. It evaluates whether a model can accurately predict the intermediate states of program execution. This is a crucial step beyond simple output prediction, as it assesses the model's ability to trace the flow of a program and understand how its internal state changes over time. Beyond these benchmarks, some research has also explored a model's understanding of code structure. For example, Ma et al. (2024a) proposed a method to evaluate LLMs by having them generate an abstract syntax tree (AST) from a given code snippet. This approach attempts to gauge a model's grasp of the hierarchical and structural relationships within code. However, as the user-provided text notes, this method faces several challenges, including a single evaluation metric, a fragmented evaluation system, and a heavy reliance on manual evaluation, which makes it difficult to scale and standardize.

## 3 CATEVAL FRAMEWORK

In this section, we first outlines task definitions from a code structure perspective and then describes in detail the two evaluation metrics: semantic accuracy rate, and structural accuracy rate. Finally,

we describe the process of building a benchmark based on the HumanEval dataset within the Code-StrcutEval framework. The fig. 1 shows an overview of our CodeStrcutEval framework.

## 3.1 TASK DEFINITION

An abstract syntax tree (AST) is a structured representation of code, generated by parsing code according to the grammar rules of a programming language. Each node of the AST corresponds to a syntactic construct or an expression, capturing the hierarchical and logical relationships inherent in the code. The AST serves to decouple the functional specification of a program (the expressions) from its computational implementation (the algorithms), thereby facilitating more precise analysis and manipulation of program structure.

To evaluate the capabilities of LLMs for code in generating and comprehending code structure, we propose a two-fold evaluation tasks: (1) Code Structure Generation (CSG), which assesses the model's ability to represent code structure (AST); and (2) Code Structure Comprehension (CSC), which evaluates the model's competence in reconstructing source code from a structured representation (AST). These complementary perspectives provide a comprehensive framework for assessing structural generation and comprehension in LLMs.

*1) Code Structure Generation (CSG):* This task involves directly generating an abstract syntax tree for a given source code. To accurately output a parseable and correct AST, LLMs must fully understand the structure of the given code, placing high demands on the model's code structure generation capability.

**Task Description**: Given a program code $C$, the model $\mathcal{M}$ aims to generate the corresponding AST, denoted as $A_{\mathcal{M}}$. The ground truth AST is represented as $A_{gt}$. This task commonly uses autoregressive code LLMs for prediction. The formal expression of this task can be illustrated as follows:

$$P_{\mathcal{M}}(A_{\mathcal{M}}) = \prod_{i=1}^{n} P_{\mathcal{M}}\left(a_i \,|\, a_{<i}, \, C, \, P_{CSG}\right) \tag{1}$$

where $P_{\mathcal{M}}(A_{\mathcal{M}})$ is the joint probability of all tokens in the predicted sequence $A_{\mathcal{M}} = (a_1, a_2, \ldots, a_n)$ by model $\mathcal{M}$. $a_i$ denotes the $i$-th node of the AST, $a_{<i}$ denotes all preceding nodes, and $P_{CSG}$ is the prompt specific to the CSG task.

*2) Code Structure Comprehension (CSC):* An abstract syntax tree (AST) represents the detailed structural information of a piece of code, and the code itself is the result of abstracting this structural information. In addition to enabling the model generate structural information from the code, we also focus on the model's ability to reverse generation the code based on this structural information. To accurately output parsable and correct code, the LLM must fully understand the structure of the given AST, but this task is much simpler than the CSG task.

**Task Description**: Given an abstract syntax tree $A$, the objective of the model $\mathcal{M}$ is to reconstruct the corresponding program code $C_{\mathcal{M}}$ by leveraging the structural information contained in $A$. The reference (ground-truth) code is denoted as $C_{gt}$. Typically, autoregressive code LLMs are employed to perform this generation task. Formally, the decoding process can be represented as:

$$C_{\mathcal{M}} = \arg\max_{C} \, P_{\mathcal{M}}(C \,|\, A, \, P_{CSC}), \tag{2}$$

where $P_{\mathcal{M}}(C \,|\, A, \, P_{CSC})$ is the conditional probability of generating code sequence $C = (c_1, c_2, \ldots, c_n)$ given the AST $A$ and the task-specific prompt $P_{CSC}$.

## 3.2 EVALUATION METRICS

*1) Semantic Accuracy Rate (SAR):* SAR measures the proportion of samples for which the generated program produces the correct output given the corresponding input. For the CSG task, the AST generated by the model is first converted into executable code, while for the CSC task, the generated

Table 1: Statistics of our benchmark dataset base HumanEval dataset.

| | CSG task | | | | CSC task | | | |
|---|---|---|---|---|---|---|---|---|
| | **Easy** | **Medium** | **Hard** | **Total** | **Easy** | **Medium** | **Hard** | **Total** |
| **Numbers** | 63 | 63 | 31 | 157 | 63 | 63 | 31 | 157 |
| **Maximum token length** | 349 | 642 | 1484 | 1484 | 60 | 118 | 245 | 245 |
| **Minimum token length** | 117 | 356 | 646 | 117 | 12 | 38 | 74 | 13 |
| **Mean token length** | 249.08 | 487.84 | 834.13 | 460.41 | 31.29 | 65.19 | 119.32 | 62.27 |

code is directly evaluated. Its mathematical definition is as follows:

$$SAR = \frac{1}{N} \sum_{i=1}^{N} \mathbb{I}\big[f(C_{\mathcal{M}}^i, U_{in}^i) = U_{out}^i\big] \tag{3}$$

where $C_{\mathcal{M}}^i$ denotes the program generated by model $\mathcal{M}$ for the $i$-th sample, $U_{in}^i$ and $U_{out}^i$ represent the input and output of the unit test, $f(\cdot, \cdot)$ denotes the function that executes the program with the given input and returns its output, and $\mathbb{I}[\cdot]$ is the indicator function.

*2) Structural Accuracy Rate (StAR):* StAR measures the structural correctness of the generated abstract syntax tree. For the CSG task, it directly evaluates the generated AST against the ground-truth AST, while for the CSC task, the generated code is first parsed into an AST before comparison. Its mathematical definition is as follows:

$$StAR = \frac{1}{N} \sum_{i=1}^{N} \mathbb{I}\big[A_{\mathcal{M}}^i \cong A_{gt}^i\big] \tag{4}$$

where $A_{\mathcal{M}}^i$ denotes the generated AST of the $i$-th sample and $A_{gt}^i$ denotes the corresponding ground-truth AST. Here, $\cong$ denotes structural equivalence between two ASTs.

### 3.3 BENCHMARK CONSTRUCTION

To construct a benchmark tailored for the CodeStructEval framework, we select the widely-recognized HumanEval dataset as our seed. HumanEval is an ideal foundation due to several critical properties. First, it comprises 164 high-quality, handwritten programming problems, each accompanied by comprehensive unit tests. This ensures that every code sample is strictly executable, which is a prerequisite for generating AST, and enables the automated calculation of correctness metrics such as SAR. Second, its manual curation process significantly mitigates the risk of data leakage, a crucial consideration for benchmarks designed to evaluate large models pre-trained on web-scale code corpora.

To adapt the seed dataset for our specific evaluation needs, we designed a multi-stage processing and filtering pipeline, which transforms the original HumanEval samples into our final benchmark. This pipeline consists of the following three stages:

1. **AST Generation.** For each problem, we parse its canonical solution to generate the corresponding AST. This process leverages Python's built-in `ast.parse()` function, providing a standardized structural representation of the source code essential for our framework's analysis.

2. **Filtering and Refinement.** To ensure the benchmark's quality and focus, we apply a two-pronged filtering strategy followed by manual review. We filter samples based on two key dimensions:

   - *Execution Time:* To maintain evaluation efficiency, we exclude samples with an execution time of 2.0 seconds or more. This step removes computationally prohibitive problems that could hinder large-scale testing.
   - *AST Complexity:* To scope the benchmark on problems of substantive but manageable complexity, we filter based on the size of the generated AST. We tokenize the AST representation using the Qwen2.5-Coder-3B tokenizer and exclude samples where the

token count exceeds 1,500. This removes extreme outliers with overly complex structures.

Finally, all remaining samples undergo a manual inspection to verify their quality and relevance.

3. **Difficulty Stratification.** For a more granular and comprehensive evaluation, we partition the refined dataset into three difficulty levels: *Easy*, *Medium*, and *Hard*. This stratification is performed by ranking each sample based on a composite analysis of its code complexity and AST structural properties. The ranked list is then partitioned into three tiers, approximating a **2:2:1** ratio for the *Easy*, *Medium*, and *Hard* categories, respectively. This approach ensures a balanced distribution of problem difficulties, allowing for a nuanced assessment of model performance.

Following this comprehensive pipeline, the final benchmark comprises 157 high-quality data instances. The detailed statistics of the resulting dataset are presented in table 1.

## 4 EXPERIMENTS

### 4.1 EXPERIMENTAL SETUP

*1) Code LLMs Selection:* To investigate the structural comprehension capability of different code LLMs, we evaluate a diverse set of models including both open-source and closed-source systems, covering multiple parameter scales, training paradigms (Base vs. Instruct), and domains (general-purpose vs. code-specialized). The selected models include Qwen2.5-Coder (Hui et al., 2024), CodeLlama (Rozière et al., 2024), StarCoder2 (Lozhkov et al., 2024), Deepseek-Coder (Guo et al., 2024), CodeGemma (Team et al., 2024), GPT-3.5-Turbo (Brown et al., 2020), GPT-4-Turbo (OpenAI et al., 2024), and Claude-3.5-Sonnet (Anthropic, 2024). For brevity, we use the following abbreviations in experiments: Qwen2.5-Coder (QC), CodeLlama (CL), StarCoder2 (SC), Deepseek-Coder (DC), and CodeGemma (CG).

*2) Parameter Settings:* We report pass@1 and pass@5 results for both SAR and StAR metrics. Unless otherwise stated, all experiments are conducted under the 1-shot setting with temperature = 0.2 and top_p = 0.9. Sampling size is fixed at 5 to ensure stable pass@K estimates. Inference is carried out on 2×A100 (40GB) GPUs using the VLLM (Kwon et al., 2023) framework. Detailed parameter configurations are provided in Appendix B.

### 4.2 PERFORMANCES OF 18 MAINSTREAM LLMS ON THE BENCHMARK

We thoroughly evaluate the code structure generation and comprehension capabilities of 18 mainstream LLMs introduced in the experimental setup subsection, using a sampling size of 1 for closed-source models and 5 for non-closed-source models. We report the pass@1 value for the two evaluation metrics: SAR and StAR. All experimental results are shown in table 2.

From table 2, we can find that: (1) The CSG task is substantially more challenging than the CSC task. For example, the SAR value of Qwen2.5-Coder-14B-Base on CSC reaches 94.52, while its performance on CSG is only 29.04. This gap indicates that translating code into its structural representation (CSG) requires precise reasoning over program semantics and syntax to construct well-formed ASTs, which is inherently more complex than generating code from a given AST (CSC), where the structural scaffold is already provided. (2) For all models and tasks, $SAR \geq StAR$, which stems from the definitions of these two evaluation metrics. SAR only evaluates the correctness of the execution results, while StAR further examines the structural alignment between generated outputs and ground-truth ASTs, imposing stricter requirements on structural consistency. (3) Within the same model series, performance generally improves as parameter size increases, consistent with the scaling law (Kaplan et al., 2020). Nevertheless, large open-source models still show limited structural reasoning ability in CSG, suggesting that scaling alone does not sufficiently enhance code structural comprehension. In contrast, closed-source models with massive parameter scales (e.g., Claude-3.5-Sonnet) demonstrate substantial advantages, particularly in tasks requiring deeper structural understanding. (4) The performance of instruct models is not always better than that of base models. For instance, in the Qwen2.5-Coder series, instruct models underperform their

Table 2: Performance of 18 mainstream code LLMs on the Benchmark dataset.

| | | CSG | | | | | | | | CSC | | | | | | | |
| | | SAR, Pass@1 (↑) | | | | StAR, Pass@1 (↑) | | | | SAR, Pass@1 (↑) | | | | StAR, Pass@1 (↑) | | | |
| | | Total | Easy | Med. | Hard | Total | Easy | Med. | Hard | Total | Easy | Med. | Hard | Total | Easy | Med. | Hard |
|---|---|---|---|---|---|---|---|---|---|---|---|---|---|---|---|---|---|
| Base | QC-1.5B | 3.69 | 9.21 | 0.00 | 0.00 | 1.66 | 4.13 | 0.00 | 0.00 | 60.76 | 70.79 | 59.05 | 43.87 | 24.59 | 27.62 | 26.35 | 14.84 |
| | QC-3B | 6.24 | 14.29 | 1.27 | 0.00 | 2.29 | 5.71 | 0.00 | 0.00 | 80.51 | 90.48 | 79.37 | 62.58 | 45.48 | 50.48 | 46.67 | 32.90 |
| | QC-7B | 23.44 | 40.32 | 17.46 | 1.29 | 12.36 | 23.17 | 7.30 | 0.65 | 89.04 | 93.02 | 89.21 | 80.65 | 64.97 | 71.43 | 64.44 | 52.90 |
| | QC-14B | 29.04 | 42.54 | 22.86 | 14.19 | 20.76 | 31.75 | 16.19 | 7.74 | 94.52 | 95.56 | 94.19 | 93.65 | 81.15 | 86.03 | 80.63 | 72.26 |
| | CL-7B | 4.97 | 12.37 | 1.27 | 0.00 | 2.42 | 6.03 | 0.00 | 0.00 | 62.55 | 71.11 | 62.86 | 44.52 | 32.61 | 38.41 | 29.21 | 27.74 |
| | CL-13B | 9.17 | 19.68 | 3.17 | 0.00 | 5.35 | 11.75 | 1.59 | 0.00 | 80.13 | 90.48 | 73.97 | 71.61 | 54.78 | 66.98 | 50.48 | 38.71 |
| | SC-7B | 4.20 | 10.48 | 0.00 | 0.00 | 1.91 | 4.76 | 0.00 | 0.00 | 75.92 | 82.22 | 79.68 | 55.48 | 52.10 | 59.37 | 54.29 | 32.90 |
| | DC-6.7B | 8.92 | 20.63 | 0.00 | 0.65 | 3.18 | 7.94 | 0.00 | 0.00 | 80.76 | 88.89 | 84.76 | 56.13 | 59.24 | 67.30 | 65.08 | 30.97 |
| | CG-7B | 7.13 | 13.97 | 3.81 | 0.00 | 2.42 | 6.03 | 0.00 | 0.00 | 84.46 | 86.67 | 85.08 | 78.71 | 58.09 | 61.27 | 61.27 | 51.61 |
| Instruct | QC-1.5B | 1.02 | 2.54 | 0.00 | 0.00 | 1.02 | 2.54 | 0.00 | 0.00 | 52.48 | 66.35 | 47.30 | 34.84 | 17.83 | 20.63 | 18.41 | 10.97 |
| | QC-3B | 3.44 | 8.57 | 0.00 | 0.00 | 1.40 | 3.49 | 0.00 | 0.00 | 81.15 | 84.44 | 80.63 | 75.48 | 41.15 | 45.08 | 44.44 | 26.45 |
| | QC-7B | 5.73 | 12.38 | 1.90 | 0.00 | 2.17 | 5.40 | 0.00 | 0.00 | 93.38 | 97.78 | 91.43 | 88.39 | 65.99 | 73.65 | 65.40 | 51.61 |
| | QC-14B | 20.76 | 25.40 | 18.41 | 16.13 | 12.36 | 19.05 | 8.39 | 7.62 | 91.97 | 93.33 | 92.06 | 89.03 | 71.72 | 75.56 | 70.79 | 65.81 |
| | CL-7B | 4.46 | 11.11 | 0.00 | 0.00 | 3.31 | 8.25 | 0.00 | 0.00 | 67.77 | 73.33 | 71.43 | 49.03 | 45.73 | 53.97 | 47.30 | 25.81 |
| | CL-13B | 8.54 | 19.05 | 2.22 | 0.00 | 6.24 | 13.97 | 1.59 | 0.00 | 76.43 | 80.95 | 80.63 | 58.71 | 57.20 | 71.75 | 55.87 | 30.32 |
| Close | GPT-3.5 | 23.57 | 31.75 | 19.05 | 16.13 | 15.92 | 28.57 | 9.52 | 3.23 | 75.16 | 76.19 | 74.60 | 74.19 | 56.05 | 57.14 | 55.56 | 54.84 |
| | GPT-4 | 70.06 | 77.78 | 69.84 | 54.84 | 57.96 | 69.84 | 55.56 | 38.71 | 97.82 | 100.00 | 96.83 | 95.24 | 78.98 | 82.54 | 80.95 | 67.74 |
| | Claude-3.5 | 91.08 | 98.41 | 87.30 | 83.87 | 79.62 | 88.89 | 79.37 | 61.29 | 99.36 | 100.00 | 100.00 | 96.77 | 98.09 | 100.00 | 98.41 | 93.55 |

base counterparts in CSG. This suggests that instruction tuning, which primarily improves general task-following ability in code generation, may not enhance and can even weaken structural reasoning ability. By contrast, CodeLlama-13B-Instruct slightly outperforms its base model in StAR on both tasks, implying that well-designed instruction data can contribute to better structural alignment. Overall, these findings highlight that current instruction tuning pipelines are insufficient to improve structural comprehension, and future work should explicitly integrate structure-aware training signals (e.g., AST-level supervision or structural constraints) to strengthen LLMs' capability in code structure understanding and generation.

## 4.3 QUANTITATIVE ANALYSIS OF CORRELATION

*1) Correlation between scores on HumanEval and our benchmark:* As powerful LLMs continue to emerge, their performances on the classic code generation benchmark HumanEval continues to break new ground. For example, the GPT-4-Turbo achieved a score of 90.2 on HumanEval. However, researchers are still curious whether these LLMs can also achieve exceptional results on other code tasks such as code comprehension. We compare the performances of 18 mainstream LLMs on our benchmark with their performances on HumanEval and plot a scatter plot, as shown in fig. 2. From this figure, we can find that: the model's scores on our benchmarks are generally positively correlated with those on HumanEval. However, a better HumanEval score doesn't necessarily mean

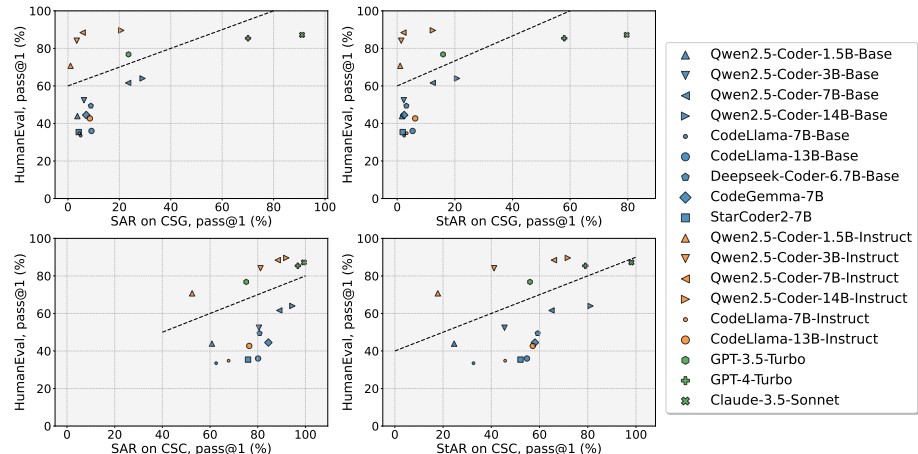

Figure 2: Scatter plot of the model's performance on our benchmark and on humaneval.

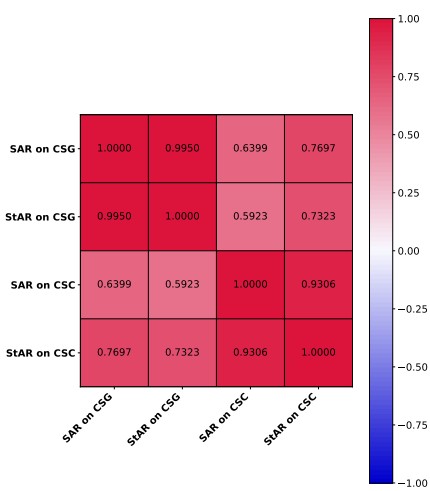

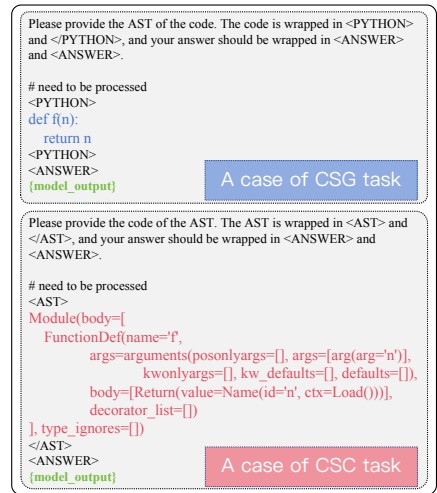

Figure 3: Correlation heatmap between CSG tasks and CSC tasks.

Figure 4: Cases of CSG and CSC tasks.

a better score on our benchmark. For example, GPT-4-Turbo's score on HumanEval is higher than Claude-3.5-Sonnet's, while Claude-3.5-Sonnet outperforms GPT-4-Turbo on our benchmark.

*2) Correlation between CSG task and CSC task:* In addition to exploring the correlation between our benchmark and HumanEval, we also explore the correlation between the CSG and CSC tasks. We calculate the Pearson coefficients of the scores of the two evaluation metrics on the two tasks based on all pass@1 values in table 2 and plot a heat map, as shown in fig. 3. From this figure, we can find that: all metrics exhibit strong positive correlations. The Pearson coefficient between the SAR and StAR scores on the CSG task was as high as 0.9950, which is the highest score.

## 4.4 HOW TO IMPROVE CODE STRUCTURE GENERATION AND COMPREHENSION CAPABILITIES?

How can we improve the model's code structure generation and comprehension capabilities? We conduct preliminary experiments using *Chain of Thought* and *Few-Shot*:

*1) Chain of Thought (CoT):* CoT is a type of prompting engineering. Its core idea is to enable LLMs to decompose complex problems into step-by-step subproblems and solve them sequentially. By explicitly outputting intermediate reasoning steps, the arithmetic, common sense, and reasoning quality of the large model are enhanced. We select a few models and report their performances with and without CoT. The experimental results are shown in table 3.

From table 3, we can find that: (1) Under the setting of pass@1, whether it is the CSG task or the CSC task, the model without CoT basically performs better than the model with CoT; (2) What is more interesting is that under the setting of pass@5, the model with CoT has a significant improvement in CSC task compared to the model without CoT. In summary, we can draw such a conclusion: the use of CoT widens the performance gap between the model in pass@1 and pass@5 on CSC task. The reason for this phenomenon is that the use of CoT gives the model more thinking space and thus more decision-making space. More decision space means that the gap between pass@1 and pass@5 will be widened. For the CSG task, since the token of the correct answer is too long, even adding CoT to the model does not significantly improve pass@5.

*2) Few-Shot:* Few-Shot technology is a learning method that allows a model to quickly adapt to a specific task by providing a small number of high-quality examples (usually 1-10). We use Qwen2.5-Coder-7B-Base, CodeLlama-7B-Base, Deepseek-Coder-7B-Base, CodeGemma-7B-Base, and StarCoder2-7B-Base to conduct a few-shot experiment, setting the number of shots to 0, 1, 2, 3, 4, 5, and 10. The results of all experiments are shown in fig. 5. From this figure, we can find that: in addition to the performance of StarCoder2-7B-Base on the CSC task, the performances of other

Table 3: Performances of models with and without CoT technology.

| | CSG, Pass@1 | | CSG, Pass@5 | | CSC, Pass@1 | | CSC, Pass@5 | |
|---|---|---|---|---|---|---|---|---|
| | **SAR** | **StAR** | **SAR** | **StAR** | **SAR** | **StAR** | **SAR** | **StAR** |
| **Qwen2.5-Coder-3B-Base** | **6.24** | **2.29** | **14.23** | **5.83** | **80.51** | **45.48** | 83.44 | 52.87 |
| **Qwen2.5-Coder-3B-Base-CoT** | 4.59 | 0.38 | 12.02 | 1.90 | 73.25 | 43.57 | **85.99** | **59.87** |
| **Qwen2.5-Coder-7B-Base** | **23.44** | **12.36** | **44.51** | **26.71** | **89.04** | **64.97** | 89.81 | 69.43 |
| **Qwen2.5-Coder-7B-Base-CoT** | 19.36 | 4.59 | 42.90 | 13.48 | 88.66 | 63.31 | **95.54** | **74.52** |
| **Qwen2.5-Coder-14B-Base** | **29.04** | **20.76** | **57.85** | **45.45** | **94.52** | **81.15** | 96.82 | 84.08 |
| **Qwen2.5-Coder-14B-Base-CoT** | 23.06 | 14.90 | 55.08 | 39.14 | 93.89 | 75.16 | **96.82** | **84.71** |
| **CodeLlama-7B-Base** | **4.97** | **2.42** | **11.68** | **5.02** | **62.55** | **32.60** | 67.52 | 39.49 |
| **CodeLlama-7B-Base-CoT** | 3.82 | 0.00 | 8.91 | 0.00 | 47.77 | 26.11 | **69.43** | **43.31** |
| **StarCoder2-7B-Base** | **4.20** | **1.91** | **10.35** | **3.78** | **75.92** | **52.10** | 78.34 | 57.32 |
| **StarCoder2-7B-Base-CoT** | 1.78 | 0.38 | 5.38 | 1.16 | 55.29 | 33.89 | **79.71** | **57.96** |
| **Deepseek-Coder-6.7B-Base** | **8.92** | **3.18** | **18.35** | **6.30** | **80.76** | **59.24** | 84.71 | 63.69 |
| **Deepseek-Coder-6.7B-Base-CoT** | 7.13 | 1.66 | 17.36 | 5.59 | 71.59 | 48.66 | **89.81** | **71.34** |
| **CodeGemma-7B-Base** | **7.13** | **2.42** | **22.13** | **7.70** | **84.46** | **58.09** | 89.81 | 72.61 |
| **CodeGemma-7B-Base-CoT** | 6.26 | 1.02 | 21.42 | 4.04 | 75.67 | 42.29 | **92.99** | **73.06** |

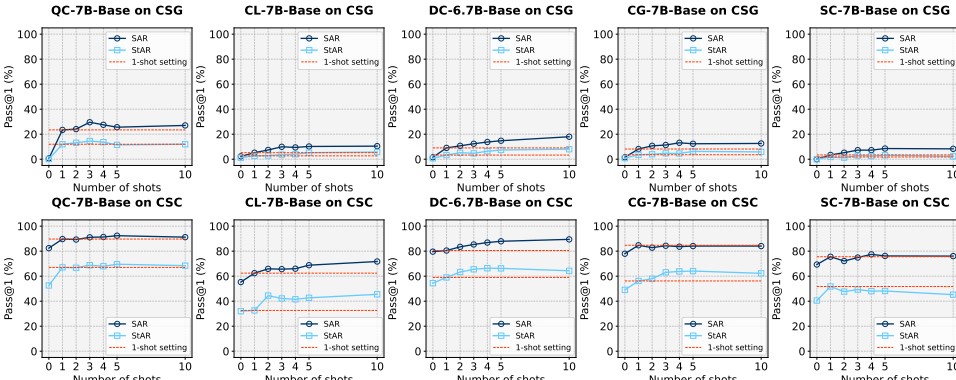

Figure 5: The impact of few-shot number on some LLMs' performances.

models on the CSG and CSC tasks increases as the number of shots increases. Among them, the CodeLlama-7B-Base has the most obvious improvement in all tasks.

# 5 CONCLUSION

We propose two tasks: Code Structure Generation (CSG) for generating abstract syntax tree (AST) from code and Code Structure Comprehension (CSC) for inferring code from abstract syntax tree (AST). Then, we construct a programming language agnostic CodeStructEval evaluation framework (comprising data preprocessing, model inference, and automated evaluation modules), and introduce the Semantic Accuracy (SAR) and Structural Accuracy (StAR) metrics. Based on the HumanEval dataset, after multiple rounds of screening and difficulty stratification, we construct a benchmark dataset containing 157 samples. Experiments on 18 mainstream LLMs show that closed-source models (such as Claude-3.5-Sonnet) outperform open-source models with small parameters. The CSG task is more difficult than the CSC task. The performance of models within the same family scales with the increase in parameters, and instruction-tuned models do not necessarily outperform the baseline model. Correlation analysis shows that the model exhibits a strong positive correlation with the HumanEval score, CSG, and CSC task metrics on this benchmark. CoT technology expand the gap between pass@1 and pass@5 on the CSC task, while Few-Shot technology improves model performance as the number of examples increases. Overall, we believe that CodeStructEval provides a complementary evaluation perspective for LLMs.

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

## A  THE USE OF LLMS

We use powerful LLMs to polish some parts of the article, and do not use LLMs otherwise.

## B  EXPERIMENTAL SETTINGS

*1) Code LLMs Selection:* To study the structural comprehension capability of code LLMs, we evaluate models with diverse characteristics: (1) **Qwen2.5-Coder** (Hui et al., 2024) (Base and Instruct, 1.5B, 3B, 7B, 14B); (2) **CodeLlama** (Rozière et al., 2024) (Base and Instruct, 7B, 13B); (3) **Star-Coder2** (Lozhkov et al., 2024) (Base, 7B); (4) **Deepseek-Coder** (Guo et al., 2024) (Base, 6.7B); (5) **CodeGemma** (Team et al., 2024) (Base, 7B); (6) **GPT-3.5-Turbo** (Brown et al., 2020); (7) **GPT-4-Turbo** (OpenAI et al., 2024); (8) **Claude-3.5-Sonnet** (Anthropic, 2024).

We mainly consider four dimensions: (i) parameter size (e.g., QC-1.5B vs. QC-14B), (ii) open-source vs. closed-source models (e.g., QC-7B vs. GPT-4-Turbo), (iii) base vs. instruction-tuned

variants (e.g., QC-7B-Base vs. QC-7B-Instruct), and (iv) general-purpose vs. code-specific models (e.g., Claude-3.5-Sonnet vs. QC-7B-Base).

*2) Parameter Settings:* We report SAR and StAR under pass@1 and pass@5. Pass@K (Chen et al., 2021) denotes the probability of obtaining at least one correct output over $K$ independent samples. For stable estimation, we set the number of samples per instance to 5. To reduce randomness and enhance reproducibility, decoding is performed with temperature $= 0.2$ and top_p $= 0.9$. Generation length is capped at 2048 tokens (extended to 4096 in the CoT setting) to prevent truncation. For efficient inference, we employ the VLLM (Kwon et al., 2023) framework with GPU_MEMORY_UTILIZATION set to 0.9, batch_size to 10, and computation in bfloat16 precision. All experiments use 1-shot prompting unless stated otherwise. The experiments are conducted on a server with 2 NVIDIA A100 GPUs (40GB each).

## C CASES IN THE EXPERIMENTS

**1) A case of Qwen2.5-Coder-7B-Base on the CSG task.**

Given a Python function. Provide the abstract syntax tree (AST) corresponding to the given Python function in the [ANSWER] and [/ANSWER] tags. Do not output any other information. The generated abstract syntax tree (AST) should be wrapped with "'ast and "', as shown in the example.
[PYTHON]
```
def f(n):
    return n
```
[/PYTHON]
[ANSWER]
```ast
Module(
    body=[
        FunctionDef(
            name='f',
            args=arguments(
                posonlyargs=[],
                args=[
                    arg(arg='n')],
                kwonlyargs=[],
                kw_defaults=[],
                defaults=[]),
            body=[
                Return(
                    value=Name(id='n', ctx=Load()))],
            decorator_list=[])],
    type_ignores=[])
```
[/ANSWER]
[PYTHON]
```
def monotonic(l: list):
    if l == sorted(l) or l == sorted(l, reverse=True):
        return True
    return False
```
[/PYTHON]
[ANSWER]
```
# model output
```ast
Module(
```

```
    body=[
        FunctionDef(
            name='monotonic',
            args=arguments(
                posonlyargs=[],
                args=[
                    arg(
                    arg='l',
                    annotation=Name(id='list', ctx=Load()))],
                kwonlyargs=[],
                kw_defaults=[],
                defaults=[]),
            body=[
                If(
                    test=BoolOp(
                    op=Or(),
                    values=[
                        Compare(
                            left=Name(id='l', ctx=Load()),
                            ops=[
                                Eq()],
                            comparators=[
                            Call(
                            func=Name(id='sorted', ctx=Load()),
                            args=[
                            Name(id='l', ctx=Load())],
                            keywords=[])]),
                        Compare(
                            left=Name(id='l', ctx=Load()),
                            ops=[
                                Eq()],
                            comparators=[
                            Call(
                            func=Name(id='sorted', ctx=Load()),
                            args=[
                                Name(id='l', ctx=Load())],
                            keywords=[
                            keyword(
                            arg='reverse',
                            value=Constant(value=True))])])]),
                    body=[
                        Return(
                            value=Constant(value=True))],
                    orelse=[]),
                Return(
                    value=Constant(value=False))],
            decorator_list=[])],
    type_ignores=[])
  # this case pass SAR and StAR tests.
```

**2) A case of Qwen2.5-Coder-7B-Base on the CSC task.**

Give an abstract syntax tree (AST) for a Python function. Please provide the Python function corresponding to the given abstract syntax tree (AST) within the [ANSWER] and [/ANSWER] tags. Do not output any additional information. The generated Python function should be wrapped with "'python and "', as shown in the example.

```
[AST]
Module(
    body=[
        FunctionDef(
            name='f',
            args=arguments(
                posonlyargs=[],
                args=[
                    arg(arg='n')],
                kwonlyargs=[],
                kw_defaults=[],
                defaults=[]),
            body=[
                Return(
                    value=Name(id='n', ctx=Load()))],
            decorator_list=[])],
    type_ignores=[])
[/AST]
[ANSWER]
```python
def f(n):
    return n
```

[/ANSWER]
[AST]
Module(
    body=[
        FunctionDef(
            name='is_sorted',
            args=arguments(
                posonlyargs=[],
                args=[
                    arg(arg='lst')],
                kwonlyargs=[],
                kw_defaults=[],
                defaults=[]),
            body=[
                Assign(
                    targets=[
                        Name(id='count_digit',
                        ctx=Store())],
                    value=Call(
                        func=Name(id='dict',
                        ctx=Load()),
                        args=[
                        ListComp(
                            elt=Tuple(
                                elts=[
                                    Name(id='i',
                                    ctx=Load()),
                                    Constant(value=0)],
                                ctx=Load()),
                            generators=[
                                comprehension(
                                    target=Name(id='i',
                                    ctx=Store()),
```

```
                                 iter=Name(id='lst',
                                 ctx=Load()),
                                 ifs=[],
                                 is_async=0)])],
                    keywords=[])),
            For(
                target=Name(id='i', ctx=Store()),
                iter=Name(id='lst', ctx=Load()),
                body=[
                    AugAssign(
                        target=Subscript(
                            value=Name(id='count_digit',
                            ctx=Load()),
                            slice=Name(id='i',
                            ctx=Load()),
                            ctx=Store()),
                        op=Add(),
                        value=Constant(value=1))],
                orelse=[]),
            If(
                test=Call(
                    func=Name(id='any', ctx=Load()),
                    args=[
                    GeneratorExp(
                        elt=Compare(
                            left=Subscript(
                            value=Name(id='count_digit',
                            ctx=Load()),
                            slice=Name(id='i',
                            ctx=Load()),
                            ctx=Load()),
                            ops=[
                                Gt()],
                            comparators=[
                                Constant(value=2)]),
                        generators=[
                        comprehension(
                        target=Name(id='i', ctx=Store()),
                        iter=Name(id='lst',
                        ctx=Load()),
                        ifs=[],
                        is_async=0)])],
                    keywords=[]),
                body=[
                    Return(
                        value=Constant(value=False))],
                orelse=[]),
            If(
            test=Call(
                func=Name(id='all', ctx=Load()),
                args=[
                GeneratorExp(
                    elt=Compare(
                        left=Subscript(
                            value=Name(id='lst',
                            ctx=Load()),
                            slice=BinOp(
```

```
                                                left=Name(id='i',
                                                ctx=Load()),
                                                op=Sub(),
                                                right=Constant(value=1)),
                                        ctx=Load())),
                                ops=[
                                    LtE()],
                                comparators=[
                                    Subscript(
                                    value=Name(id='lst',
                                    ctx=Load()),
                                    slice=Name(id='i',
                                    ctx=Load()),
                                    ctx=Load())]),
                            generators=[
                                comprehension(
                                    target=Name(id='i',
                                    ctx=Store()),
                                    iter=Call(
                                    func=Name(id='range',
                                    ctx=Load()),
                                    args=[
                                        Constant(value=1),
                                        Call(
                                        func=Name(id='len',
                                        ctx=Load()),
                                        args=[
                                        Name(id='lst',
                                        ctx=Load())],
                                        keywords=[])],
                                        keywords=[]),
                                        ifs=[],
                                        is_async=0)])],
                        keywords=[]),
                        body=[
                            Return(
                                value=Constant(value=True))],
                        orelse=[
                            Return(
                                value=Constant(value=False))])],
                decorator_list=[])],
        type_ignores=[])
[/AST]
[ANSWER]
# model output
def is_sorted(lst):
    count_digit = dict([(i, 0) for i in lst])
    for i in lst:
        count_digit[i]+=1
    if any(count_digit[i] > 2 for i in lst):
        return False
    if all(lst[i-1] <= lst[i] for i in range(1, len(lst))):
        return True
    else:
        return False
# this case pass SAR and StAR tests.
```

**3) The prompt of CSG task with CoT.**

> Given a Python function. Please think step by step and provide the abstract syntax tree (AST) corresponding to the given Python function in the [ANSWER] and [/ANSWER] tags. Do not output any other information. The generated abstract syntax tree (AST) should be wrapped with "'ast and "', as shown in the example.
> [PYTHON]
>
> ```
> def f(n):
>     return n
> ```
>
> [/PYTHON]
> [THOUGHT]
> Let's think about it step by step.
> 1. To generate the AST representation, I start by analyzing the given Python function. The function is a module-level definition, so the root node is Module.
> 2. The Module's body contains a single FunctionDef node since there's one function definition. The FunctionDef has the name 'f' as specified.
> 3. Next, the arguments for the function: it takes one positional argument 'n', so the arguments node has posonlyargs and kwonlyargs as empty lists, args contains a single arg node with arg='n', and defaults and kw_defaults are empty.
> 4. The body of the function is a single Return statement. The return value is the Name 'n' with context Load, as it's being referenced.
> 5. Decorator_list is empty since there are no decorators, and type_ignores in the Module is also empty. Putting all these elements together gives the AST structure.
> [/THOUGHT]
> [ANSWER]
>
> ```ast
> Module(
>     body=[
>         FunctionDef(
>             name='f',
>             args=arguments(
>                 posonlyargs=[],
>                 args=[
>                     arg(arg='n')],
>                 kwonlyargs=[],
>                 kw_defaults=[],
>                 defaults=[]),
>             body=[
>                 Return(
>                     value=Name(id='n', ctx=Load()))],
>             decorator_list=[])],
>     type_ignores=[])
> ```
>
> [/ANSWER]
> [PYTHON]
>
> ```
> {python_code}
> ```
>
> [/PYTHON]
> [THOUGHT]
>
> ```
> {model_thought}
> ```
>
> [/THOUGHT]
> [ANSWER]
>
> ```
> {model_answer}
> ```
>
> [/ANSWER]

**4) The prompt of CSC task with CoT.**

Give an abstract syntax tree (AST) for a Python function. Please think step by step and provide the Python function corresponding to the given abstract syntax tree (AST) within the [ANSWER] and [/ANSWER] tags. Do not output any additional information. The generated Python function should be wrapped with "'python and "', as shown in the example.
[AST]

```
Module(
    body=[
        FunctionDef(
            name='f',
            args=arguments(
                posonlyargs=[],
                args=[
                    arg(arg='n')],
                kwonlyargs=[],
                kw_defaults=[],
                defaults=[]),
            body=[
                Return(
                    value=Name(id='n', ctx=Load()))],
            decorator_list=[])],
    type_ignores=[])
```

[/AST]
[THOUGHT]
Let's think about it step by step.
1. I need to convert the given AST into a Python function. First, I look at the top-level node, which is a Module. The body of the Module contains a single FunctionDef. The FunctionDef has the name 'f', so the function is named f.
2. Next, the args of the FunctionDef is an arguments object. The args list within arguments has one arg with name 'n', so the function takes one parameter 'n'. There are no posonlyargs, kwonlyargs, or defaults, so it's a simple parameter list.
3. The body of the FunctionDef has a single Return statement. The value of the Return is a Name with id 'n' and Load context, meaning it's returning the parameter 'n'.
4. Putting this together, the function is defined as def f(n): with a return statement that returns n.
[/THOUGHT]
[ANSWER]

```python
def f(n):
    return n
```

[/ANSWER]
[AST]

{AST}

[/AST]
[THOUGHT]

{model_thought}

[/THOUGHT]
[ANSWER]

{model_answer}

[/ANSWER]

