# OpenReview forum: "CodeStructEval: A Holistic Evaluation Framework of Code Structure Generation and Comprehension"
_ICLR.cc/2026/Conference — Submitted to ICLR 2026_

### Official Review · Reviewer_9Gcy · 2025-10-27

**Soundness:** 3
**Presentation:** 4
**Contribution:** 1
**Rating:** 0
**Confidence:** 5

**Summary:**

This paper explores a new sub-direction in the evaluation of code generation capabilities for large language models (LLMs), focusing on code structure generation and comprehension, which have been previously unexplored. The authors define two tasks, CSC and CSG, targeting code structure generation and code structure comprehension, respectively. They introduce the CodeStructEval framework and two evaluation metrics (SAR and StAR) for assessing accuracy. A benchmark is collected to support research on this topic. The paper also reviews results from 18 mainstream LLMs within the proposed evaluation framework, which serve as baselines and highlight the need for further research on this sub-direction.

**Strengths:**

Originality: Unlike previous benchmarks that primarily evaluate the ability of LLMs to generate sequential code based on natural language requirements, this paper proposes an assessment of their capability in generating and understanding code structure, a new sub-direction in code generation capabilities evaluation.

Quality: The paper selects the widely recognized HumanEval dataset as the seed data. Its manually curated process significantly reduces the risk of data leakage, which is a crucial consideration for benchmarks designed to evaluate large models pre-trained on web-scale codebases.

Clarity: Clear and concise writing style and tables. The charts are clear and easy to understand.

Significance: Provides a strong method for evaluating code-structure-understanding abilities. It may help evaluate code-logic-understanding abilities.

**Weaknesses:**

Nowadays, people expect code LLMs to generate entire projects. So the ability of LLMs to generate projects should be the focus of new benchmarks. This capability includes at least three core competencies: the ability to acquire domain knowledge, the ability to understand human intent and engage in multi-turn conversations, and the LLM's inherent logical and reasoning abilities. Among these, the ability to acquire domain knowledge is the most important. Projects often rely heavily on third-party libraries. These libraries may not have been included in the LLM's training data. An LLM's ability to learn to use these third-party libraries is crucial for generating correct project code. However, the benchmark proposed in this paper fails to cover any of the three core competencies convincingly.

First, the benchmark does not test the LLM's ability to acquire domain knowledge. HumanEval does not rely on third-party libraries. All its problems can be solved using only the Python standard library. Explanations of code ASTs are also available in Python's documentation. LLMs have likely memorized these explanations thoroughly during pre-training.

Second, the benchmark does not assess the LLM's ability to understand human intent or engage in multi-turn conversations. This deviates significantly from real-world scenarios, where humans use LLMs for development.

Third, the benchmark may evaluate an LLM's logical and reasoning abilities. But many existing benchmarks already assess these skills, such as those centered on mathematical problems. The paper does not demonstrate how its proposed benchmark outperforms these existing ones. Even if it claims that converting between code and its AST is a unique, coding-critical logical or reasoning skill, it provides no evidence to support this claim.

As a result, LLM scores from the paper's benchmark may not match humans' experience using these LLMs for coding tasks.

In addition, the experiments rely on relatively outdated models, which limit the relevance to current LLMs. Some cutting-edge LLMs were not included in the tests.

**Questions:**

1. Although line 053 of the paper states that "existing research shows that providing code structure information during model training can effectively improve the model's performance in various downstream tasks," I remain unconvinced by the claim in line 018 that "The two aspects represent a deeper, more fundamental understanding of program logic, better reflecting the model's capabilities for logical reasoning and structural awareness." Specifically, I question whether evaluating a model's program logic reasoning ability necessarily depends on analyzing code structure. Can you provide concrete evidence to support this argument?

2. The paper would be significantly strengthened by additional experiments evaluating more recent LLMs, especially those with reasoning abilities. Would it be possible for the authors to include such experiments in the revision?

3. The paper does not clarify how many of the wrong answers are due to format or syntax errors rather than misunderstandings of code structure. Would it be possible for the authors to report the proportion of different error types or at least share raw experiment data?

4. It would help clarity and reproducibility to specify the exact release dates and versions of the models (for example, gpt-4-0613). Could the authors provide such information?

5. Typo on line 053: addtion -> addition.

---

### Official Review · Reviewer_a2di · 2025-10-27

**Soundness:** 2
**Presentation:** 2
**Contribution:** 1
**Rating:** 2
**Confidence:** 4

**Summary:**

The paper proposes CodeStructEval, a framework to test large language models on two code–structure tasks: Code Structure Generation (producing an AST from code) and Code Structure Comprehension (reconstructing code from an AST). It defines two metrics—Semantic Accuracy Rate (execution correctness) and Structural Accuracy Rate (AST equivalence), and builds a benchmark of 157 HumanEval-derived items stratified by difficulty. Using this setup, the authors report results for 18 LLMs.

**Strengths:**

- Widely select LLMs. This paper selected a diverse set of models, including both open-source and closed-source systems and multiple model families. Although 30B-level or 70B-level models were not evaluated, the intensity of the evaluation is still worth affirming.

- Clear writing. The paper clearly introduces the approach and experiments. Both the figures and paragraphs are easy to follow.

**Weaknesses:**

Prior studies have formulated code structure understanding tasks to evaluate LLMs.
For example, previous work [1] formulated the *AST generation* task.
Besides, previous work [2] explored the code pre-trained models' abilities on both syntax and semantics structure prediction or tagging. Their *syntax node pair prediction* and *token syntax tagging* tasks tested models' understanding of AST structure.
Therefore, the present work should carefully articulate and demonstrate its novelty or advantages relative to these prior efforts.

The authors claimed that one of the advantages of the previous work [1] is that the previous work used a manual evaluation. However, [1] used manual evaluation for a better evaluation to eliminate the impact of minor issues (missing trivial leaf nodes) or AST string formats (they only refer to the tree-sitter format, but do not strictly compare).
However, this work did not implement an automatic evaluation with similar effectiveness. Instead, this work ignored issues that [1] solved by manual evaluation. Further explanations are needed on whether this work used exact matching and how to deal with minor issues/format gaps.

The LLMs are generally not required to generate the string-serialized AST.
Different AST parsers also have different formats of string serializations. For example, the cases in Figure 4 show a different format from [tree-sitter AST](https://github.com/tree-sitter/py-tree-sitter/blob/1d87ce7b3e385b92ea5874d1bbd506f8602b8a14/README.md?plain=1#L141-L151).
In this work's task design, the prompt does not explain the AST string format, nor does it mention the AST library they used.
Therefore, the task is highly reliant on LLM's familiarity with the specified serialized AST format, but not on a general ability of code structure generation or comprehension, as the authors claimed.
This design could be improved to have a more reasonable evaluation.

Only HumanEval was used in this work, which is a well-studied and small benchmark. The reference value of this work can be further increased by including larger-scale, multilingual, or repository-level datasets.

Generally, the AST and source code should have a one-to-one mapping. Each task has its exact ground truth. Therefore, the value of using a test suite for evaluation is questionable.

The results part provides counterintuitive data that people would be concerned about.
For example, in Figure 2, CSG results from many models are close to zero, while the code generation abilities shown by the HumanEval are very diverse. In Table 3, the model without CoT basically performs better than the model with CoT.
It would be better if these could be further explained.

---

1. Ma, Wei, et al. "Lms: Understanding code syntax and semantics for code analysis." arXiv preprint arXiv:2305.12138 (2023).
2. Ma, Wei, et al. "Unveiling code pre-trained models: Investigating syntax and semantics capacities." ACM Transactions on Software Engineering and Methodology 33.7 (2024): 1-29.

**Questions:**

What does the symbol $\cong$ represent in Equation (4)? It appears to indicate strict equality. If an advanced algorithm was implemented to determine AST equivalence, please describe it in detail.

---

### Official Review · Reviewer_7s4o · 2025-10-30

**Soundness:** 1
**Presentation:** 3
**Contribution:** 2
**Rating:** 2
**Confidence:** 4

**Summary:**

This paper proposes to evaluate the code generation capabilities of LLMs via a novel method: generating (CSG = code->AST) and comprehending (CSC = AST -> code) code structures. The structure of choice is ASTs. They propose two tasks (CSG and CSC) and two metrics (SAR = semantic via unit tests, and StAR structural AST equivalence). Using a popular benchmark, HumanEval, and some human effort, they create 157 examples and evaluate 18 LLMs. This is framed as a new benchmark, CodeStructEval. Naturally they find that LLMs can generate correct code from ASTs (CSC is easier than CSG) much easier than they can generate a desired AST from a given source program and that LLMs underperform one of the new metrics, StAR.

**Strengths:**

* 2 metrics are proposed which complement each other: one that measures the claim they are after (StAR) and one that measures the original mainstream goal (SAR).
* The authors make a good effort to evaluate many (18) models across different types like open vs closed source, and code specific vs general purpose models.
* The authors offer some brief takeaways given their extensive empirical results.

**Weaknesses:**

* The paper claims that CodeStructEval is language-agnostic, however, all code is derived from python. Worse, is that the paper never gives a formal definition of an Abstract Syntax Tree (AST). Instead, on line 259 they mention that the correct AST is derived from, “Python’s built-in ast.parse”. There is no version of python pinned and no discussion of the assumptions that the function assumes when creating an AST.
The choice of ASTs for code structure is arbitrary. The authors offer no ablations or discussion on why ASTs are a superior choice for code structure over alternative structures such as Control Flow Graphs, Data Flow Graphs, etc.
* Related work is a timeline at best and fails to actually identify works that look at code representations or even more generally, generating intermediate representations which are then used to generate the final programs.
  * A quick search finds, “SAGE-HLS: Syntax-Aware AST-Guided LLM for High-Level Synthesis Code Generation,” by Khan et al which would at least invalidate the novelty claim of this paper as Khan et al. generate ASTs amongst other strategies to generate a target language (Verilog) of choice.
   * Other examples of intermediate representations (again the feedback here is that this entire area of research is not even acknowledged rather than these specific examples): ​​Can Large Language Models Understand Intermediate Representations by Jiang et al. and ComPile: A Large IR Dataset from Production Sources by Grossman et al.
* The definition of structural equivalence is completely underspecified.

**Questions:**

* In section 4.4 you make an arbitrary choice of the evaluations techniques: CoT and Few-Shot. Other than those being popular, why did you not mention techniques like constrained decoding since you already have ASTs (one could imagine something like PICARD by Scholak)?
* Same section, third paragraph you find that under pass@1 CoT does WORSE but then at pass@5 CoT does BETTER. These are contradicting results which is fine, however, your conclusion on line 423 is, “The reason for this phenomenon is that the use of CoT gives the model more thinking space and thus more decision-making space”. How does this explain the contradicting phenomenon?
* On line 425, you say, “For the CSG task, since the token of the correct answer is too long, even adding CoT to the model does not significantly improve pass@5”. This is unclear to me. Is this because you didn’t sample 5 times under pass@5 and asked for all 5 in the same prompt or because your generation length was set to 4096? The former seems like an incorrect experiment while the latter seems like you should have repeated the experiment under better conditions.
* StAR is not reproducible from the paper as is. What exactly is structural equivalence and how is the distance measured?
* The paper does not offer justification as to why StAR is a good metric to use. Consider the goal of writing an in-place sorting algorithm. Two acceptable solutions are 1) a recursive, stable merge sort, and 2) an interactive shell sort with a gap sequence like Ciura. The ASTs produced by python’s ast.parse() are completely different but both are functionally and semantically correct. Thus, what is the motivation for using StAR over unit tests that exist in benchmarks like HumanEval? Empirical evidence and theoretical intuition would be nice. FWIW, I know that program equivalence is an undecidable problem and so I am open to new metrics, however, they should be well motivated.

---

### Official Review · Reviewer_F6ik · 2025-10-31

**Soundness:** 3
**Presentation:** 3
**Contribution:** 2
**Rating:** 4
**Confidence:** 4

**Summary:**

This work propose CodeStructEval, a benchmark to assess the LLMs' capabilities in Code Structure Generation (CSG): generating abstract syntax tree (AST) and Code Structure Comprehension (CSC): generating code from AST. CodeStructEval was constructed from HumanEval and consisted of 157 high-quality samples with difficulties of easy, medium, and hard. The evaluation reports various LLMs on CodeStructEval and include several analyses.

**Strengths:**

- Both CSG and CSC tasks are important for code comprehension.
- The dataset is high quality and the experiments involve several additional analyses.

**Weaknesses:**

- My main concerns is that the CodeStructEval benchmark seems to be too easy for close-sourced LLMs. Old models such as GPT-4 and Claude-3.5 already achieved very high performances in many tasks, and even near-perfect scores on CSC. It is likely that newer models like Clade 4 and GPT-5 can achieve perfect scores on this benchmark, which gives little room for future development. It is better to include newer candidates such as Grok 4, Llama4, Deepseek V3, (3.1 and 3.2), etc.
- Since benchmarking LLMs on code understanding tasks is quite saturated, it is better to also evaluate agentic solutions.
- The scale of this benchmark is quite small, with only 157 samples, it is difficult to cover many code scenarios such as programming languages, coding tasks, etc.

**Questions:**

See weaknesses.

---

### Meta-Review · Area_Chair_xgPB · 2026-01-08

**Summary:**

The reviewers have raised significant, multifaceted concerns across all evaluation criteria, leading to a majority recommendation for Rejection： two rejections (Reviewer 7s4o and a2di), one strong rejection (Reviewer 9Gcy), and one marginally below the acceptance threshold (Reviewer F6ik). The authors do not submit any responses during the rebuttal period, so all of the concerns raised by the reviewers are not addressed.

The primary contribution of this paper is a new benchmark focusing on Code Structure Generation (CSG: Code  AST) and Code Structure Comprehension (CSC: AST  Code)—is criticized for its limited novelty (Reviewers cite prior work on AST generation and intermediate representations that are not adequately acknowledged) and its small scale (157 samples, based solely on Python's HumanEval). The core methodology is undermined by a lack of formal specification for the AST structure used and the Structural Accuracy Rate (StAR) metric, making the work non-reproducible as currently described. Specifically, the reliance on a single, unspecified Python AST parser format makes the task an evaluation of an LLM's familiarity with a specific serialization format rather than a general ability for structure understanding. Furthermore, the motivation for using StAR over existing semantic metrics (unit tests) is questioned, especially since structurally different, yet functionally equivalent, programs are common. Finally, the experiments are deemed outdated, lacking evaluation of current state-of-the-art LLMs, and the reported counterintuitive results (e.g., CoT performing worse at pass@1, poor CSG performance, high CSC performance on older models) are not convincingly explained. The consensus indicates that while the presentation is generally clear and the topic of code structure understanding is important, the paper is insufficient for acceptance at this time.

**Reviewer Concerns:**

N/A. The authors do not submit any responses during the rebuttal period, so all of the concerns raised by the reviewers are not addressed.

**Reviewer Scores:**

N/A. The authors do not submit any responses during the rebuttal period, so all of the concerns raised by the reviewers are not addressed.

---

### Decision · Program_Chairs · 2026-01-26

Reject